# The Clinical Utility of Dual-Energy Computed Tomography in the Diagnosis of Gout—A Cross-Sectional Study

**DOI:** 10.3390/jcm11175249

**Published:** 2022-09-05

**Authors:** Maria Sotniczuk, Anna Nowakowska-Płaza, Jakub Wroński, Małgorzata Wisłowska, Iwona Sudoł-Szopińska

**Affiliations:** 1Department of Radiology, National Institute of Geriatrics, Rheumatology and Rehabilitation, 02-637 Warsaw, Poland; 2Department of Rheumatology, National Institute of Geriatrics, Rheumatology and Rehabilitation, 02-637 Warsaw, Poland

**Keywords:** gout, dual-energy computed tomography, monosodium urate crystals

## Abstract

Dual-energy computed tomography (DECT) is an imaging technique that detects monosodium urate (MSU) deposits. This study aimed to assess the clinical utility of DECT in the diagnosis of gout. A total of 120 patients with clinical suspicion of gout who underwent DECT were retrospectively enrolled. The sensitivity and specificity of DECT alone, American College of Rheumatology (ACR)/European Alliance of Associations for Rheumatology (EULAR) classification criteria without DECT, and ACR/EULAR criteria with DECT were assessed. Additionally, an analysis of gout risk factors was performed. When artifacts were excluded, any MSU volume provided the best diagnostic value of DECT (AUC = 0.872, 95% CI 0.806–0.938). DECT alone had a sensitivity of 90.4% and specificity of 74.5%. Although ACR/EULAR criteria without DECT provided better diagnostic accuracy than DECT alone (AUC = 0.926, 95% CI 0.878–0.974), the best value was obtained when combing both (AUC = 0.957, 95% CI 0.924–0.991), with 100% sensitivity and 76.6% specificity. In univariate analysis, risk factors for gout were male sex, presence of tophi, presence of MSU deposits on DECT, increased uric acid in serum (each *p* < 0.001), and decreased glomerular filtration rate (GFR) (*p* = 0.029). After logistic regression, only increased serum uric acid (*p* = 0.034) and decreased GFR (*p* = 0.018) remained independent risk factors for gout. Our results suggest that DECT significantly increases the sensitivity of the ACR/EULAR criteria in the diagnosis of gout.

## 1. Introduction

Gout is the most common inflammatory arthritis, and its prevalence is increasing in Western societies [1]. The inflammation is caused by the deposition of monosodium urate (MSU) crystals in and around joints [2]. An increased risk of developing gout is associated with male sex, obesity, hyperuricemia, and a diet rich in purines [3]. A gout attack presents as a sudden, painful swelling of the joint, usually the first metatarsophalangeal or ankle joint. Chronic gout leads to joint destruction and the formation of tophi in soft tissues [1,2].

The gold standard for the diagnosis of gout is a visualization of the presence of negatively birefringent MSU crystals under polarizing microscopy in a sample of aspirated synovial fluid from an affected joint. However, this is an invasive procedure and can be difficult to perform, especially in small joints. Moreover, false negative results can occur due to low concentrations of crystals in the early stage of the disease [2]. The search for MSU crystals can also be negative in the case of extra-articular gout involvement.

The 2015 American College of Rheumatology (ACR)/European Alliance of Associations for Rheumatology (EULAR) gout classification criteria are often used in clinical trials [4]. This scoring system includes clinical criteria and laboratory and imaging findings. Imaging modalities that can facilitate the diagnosis of gout and are included in ACR/EULAR criteria are plain radiography (presence of typical bone erosions), ultrasound (presence of a “double contour” sign), and dual-energy computed tomography (DECT).

DECT, which has become more popular over the last few years, enables the detection and quantification of MSU crystal deposition in joints, tendons, and periarticular soft tissue. DECT does not require a contrast agent; instead, it uses tissue-specific attenuation. Data are acquired at 80 kV and 140 kV and analyzed using a two-material decomposition algorithm designed for gout that color-codes urate [4,5,6]. A positive scan is defined as the presence of color-coded MSU depositions at articular or periarticular sites. Nail-bed, submillimeter-sized, skin, motion, menisci, costal cartilage, and vessels should be excluded as artifacts [4,7,8]. DECT has a sensitivity of 79–93% and a specificity of 75–90% [9,10,11,12]. Nevertheless, false-negative results are considered common in patients with a recent onset of gout [13,14].

The aim of this study was to assess the diagnostic value of DECT in patients with clinical suspicion of gout. Additionally, an analysis of gout risk factors was performed.

## 2. Materials and Methods

Patients who were hospitalized in the reference center for rheumatic disease from January 2018 until February 2021 due to clinical suspicion of gout and underwent DECT were retrospectively enrolled in the study. The study was approved by the institutional ethics board (KBT-1/2/2022). Patients with no medical history available (electronic or paper) were excluded from the study.

The following data were also collected: patient characteristics (age, sex, and BMI), comorbidities (hypertension, diabetes, chronic kidney disease, kidney stones, psoriasis, hypothyroidism, dyslipidemia, obesity [defined as BMI ≥ 30 kg/m^2^], and inflammatory rheumatic conditions, such as rheumatoid, psoriatic, and unspecified arthritis, other spondyloarthropathies, and connective tissue diseases), laboratory findings (serum uric acid level, uric acid in urine, 24 h urine collection, creatinine and glomerular filtration rate (GFR), C-reactive protein, and full lipid profile), presence of tophi, and history of urate-lowering therapy. Data from imaging findings included in the database were: the presence of typical erosions on X-ray and MSU deposits on DECT (with their location and volume).

For DECT, a dual-energy scanner (Siemens Somatom Definition AS 128-slices) with voltage of 80/140 kV was used. All examinations were reconstructed using bone algorithm 0.75 mm slices. Software program syngo.via 4.6.6 Siemens Healthineers (Warsaw, Poland) was used for post-processing with a “gout” preset. The MSU crystals were color-coded green. DECT was considered positive based on the automatically calculated result checked by two experienced radiologists who excluded possible artifacts. The DECT analysis was performed by radiologists blinded to clinical and laboratory findings.

Due to the lack of synovial fluid examination in all patients, in our study, we used clinical diagnosis of gout as the diagnostic gold standard. The clinical diagnosis of gout was made by two rheumatologists (attending physician and independent expert) based on patients’ symptoms, history of gout risk factors and comorbidities, physical examination, and laboratory and imagining findings. Additionally, all patients were retrospectively analyzed using the 2015 ACR/EULAR gout classification criteria—both with and without taking into account the DECT results.

### Statistical Analysis

For each of the diagnostic methods (DECT alone, ACR/EULAR classification criteria without DECT, and ACR/EULAR classification criteria with DECT), the sensitivity and specificity were calculated, and the receiver operating characteristic (ROC) curves were obtained with the calculation of the area under the curve (AUC). The compliance of the data with the normal distribution was assessed using the Shapiro–Wilk test. The significance of the observed differences between the two groups was assessed using the Student’s *t*-test for variables with a normal distribution, the Mann–Whitney U test for variables without a normal distribution, and the chi-square test or Fisher’s exact test (for tables with values less than 5) for categorical variables. In multivariate analysis, logistic regression analysis was used. Statistical significance was set at *p* < 0.05. Statistical analysis was performed using Statistica 13.3 software (StatSoft Polska, Kraków, Poland).

## 3. Results

A total of 120 patients (84 men and 36 women) were enrolled in the study. Out of them, 88 (73.3%) patients had more than one anatomical area scanned with DECT. Overall, 318 anatomical areas were examined, and 180 (57%) were positive for MSU crystals on DECT (Table 1).

Feet and ankles, followed by knees, were the most common sites for MSU depositions. Examples of DECT scans positive for MSU crystals in a foot and ankle and in a knee are presented in Figure 1 and Figure 2, respectively.

A total of 73 (58%) patients were finally clinically diagnosed with gout. Patient characteristics are presented in Table 2, and gout diagnostic features are presented in Table 3.

DECT detected MSU deposits in 96 (80%) patients. The optimal cut-off point for the MSU volume was calculated to be 0.05 cm^3^ (AUC = 0.779, 95% CI 0.692–0.866). However, after radiological assessment, 18 positive DECT results were evaluated as artifacts (mainly nails and menisci). Examples of artifacts are presented in Figure 3.

The best diagnostic value of DECT, after artifacts exclusion, was obtained with any MSU volume (AUC = 0.872, 95% CI 0.806–0.938), with a DECT sensitivity of 90.4% and specificity of 74.5%.

The 2015 ACR/EULAR gout classification criteria without DECT provided better diagnostic accuracy than DECT alone, with 74% sensitivity and 91.5% specificity (AUC = 0.926, 95% CI 0.878–0.974). The best diagnostic value was obtained with the ACR/EULAR criteria taking into account DECT results, with 100% sensitivity and 76.6% specificity (AUC = 0.957, 95% CI 0.924–0.991). In the studied group, DECT enabled a gout diagnosis in 19 additional patients and the exclusion of gout in 3 patients. ROC curves of the gout diagnostic tools are presented in Figure 4.

The following risk factors for gout were identified in a univariate analysis: male sex, presence of tophi, presence of MSU deposits on DECT, increased uric acid in serum (each *p* < 0.001), and decreased GFR (*p* = 0.029). After logistic regression, only increased serum uric acid (*p* = 0.034) and decreased GFR (*p* = 0.018) remained independent risk factors for gout.

## 4. Discussion

This cross-sectional study showed that the detection of MSU crystals on DECT can significantly improve gout diagnosis. We found that DECT alone has a sensitivity of 90.4% and a specificity of 74.5%, which is consistent with the results of previous studies [9,10,11,12]. However, DECT should not be used alone without consideration of other gout features. In our study, the best diagnostic accuracy was obtained with the 2015 ACR/EULAR gout classification criteria including DECT, with 100% sensitivity and 76.6% specificity. This is in line with a previous study by Gamala et al., who found that DECT has an additive value in ACR/EULAR gout classification criteria [15].

To the best of our knowledge, this is the first study to assess the relationship between the volume of MSU deposits on DECT and the diagnosis of gout. After DECT artifacts were excluded, any positive MSU result provides the best diagnostic value, regardless of the size of the deposit. However, if there is no possibility of the manual exclusion of artifacts by the radiologist, the cut-off value of 0.05 cm^3^ provides the best diagnostic value.

The artifact identification is executed subjectively and requires a large amount of experience. In our study, the most challenging step was the differentiation of small amounts of MSU deposits (sometimes as little as 0.01 cm^3^) from artifacts. This could give false-positive results. Although there are studies that have attempted to optimize DECT post-processing settings to reduce artifacts, additional studies are needed to improve the automatic elimination of all artifacts [6,7,8,16,17,18,19].

Our study also showed the need for more reliable gout diagnosis in patients with suspected gout. Primary care physicians often prescribe uric-acid-lowering drugs to patients with suspected gout due to hyperuricemia associated with arthralgia. In this study, 11 (23.4%) patients without gout were taking urate-lowering treatment. It cannot be unequivocally assessed whether this was due to an overtreatment or observation of patients in remission due to the dissolution of MSU crystals after uric-acid-lowering treatment. The use of the 2015 ACR/EULAR criteria in conjunction with DECT resulted in drug withdrawal in these patients. Other inflammatory joint diseases should be taken into account in the differential diagnosis. We found significantly more autoimmune arthritis among patients in whom gout was excluded than in patients with confirmed gout (29.8% vs. 9.6%) (Table 2).

Risk factors for gout are well-studied and include male sex, obesity, chronic kidney disease, hypertension, diabetes, and hyperuricemia [20,21]. In our study, only increased serum uric acid (*p* = 0.034) and decreased GFR (*p* = 0.018) remained independent risk factors for gout in multivariate analysis, probably due to the small sample size.

Our study has several other limitations. The first limitation is the lack of data on the onset of symptoms. We obtained some false-negative DECT results, which are likely due to the examination taking place at an early disease stage, when the MSU crystals did not have time to be deposited in the joints [13]. Secondly, ultrasound results were not included in our analysis. Finally, the major limitation is the fact that none of the patients had synovial fluid tested for MSU crystals. Although this is theoretically the gold standard for gout diagnosis, the synovial fluid aspiration procedure is invasive and technically difficult to perform in the joints most commonly affected by gout, and false negative results can occur due to low concentrations of crystals in the early stage of the disease. Therefore, gout in everyday clinical practice is diagnosed clinically.

## 5. Conclusions

DECT proved to be a helpful tool in the diagnosis of gout in a real-world setting, significantly increasing the sensitivity of the 2015 ACR/EULAR criteria. DECT could therefore be beneficial for patients with suspected gout by providing an earlier diagnosis and treatment. Both clinicians and radiologists should be aware of possible artifacts that can lead to false-positive results. As such, DECT should not be used alone as a diagnostic tool.

## Figures and Tables

**Figure 1 jcm-11-05249-f001:**
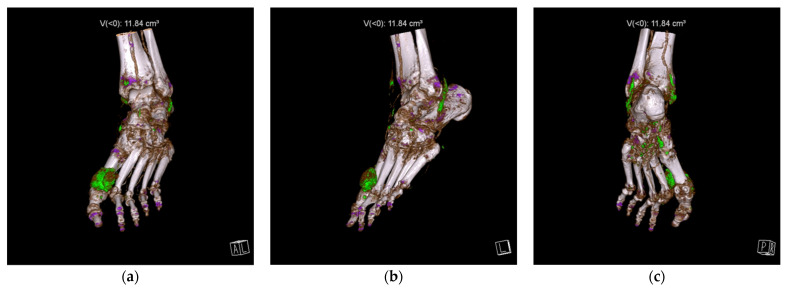
Dual-energy computed tomography 3D reconstruction of a foot positive for monosodium urate (MSU) crystals (color-coded green) in anterior (**a**), lateral (**b**), and posterior (**c**) views. The MSU deposits are present around the first metatarsophalangeal joint (**a**,**b**) and around multiple tendons in the ankle and foot (**b**,**c**). Volume of the MSU deposits was automatically calculated (11.84 cm^3^).

**Figure 2 jcm-11-05249-f002:**
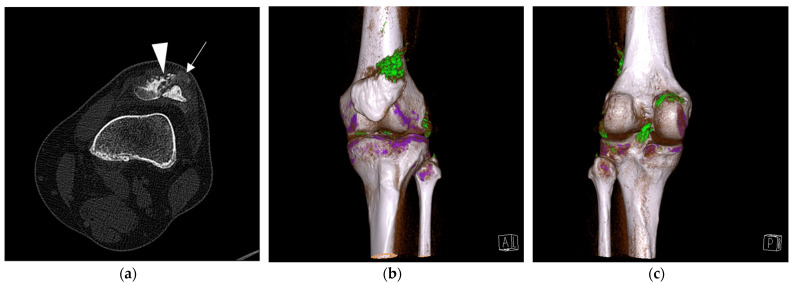
Gout in a knee. (**a**) Computed tomography scan shows erosions (arrowhead) and a possible tophus (arrow) at the lateral aspect of the patella base. (**b**) Dual-energy computed tomography 3D reconstruction confirms monosodium urate (MSU) deposits (color-coded green) in this location. (**c**) Dual-energy computed tomography 3D reconstruction shows MSU deposits in the posterior compartment of the knee.

**Figure 3 jcm-11-05249-f003:**
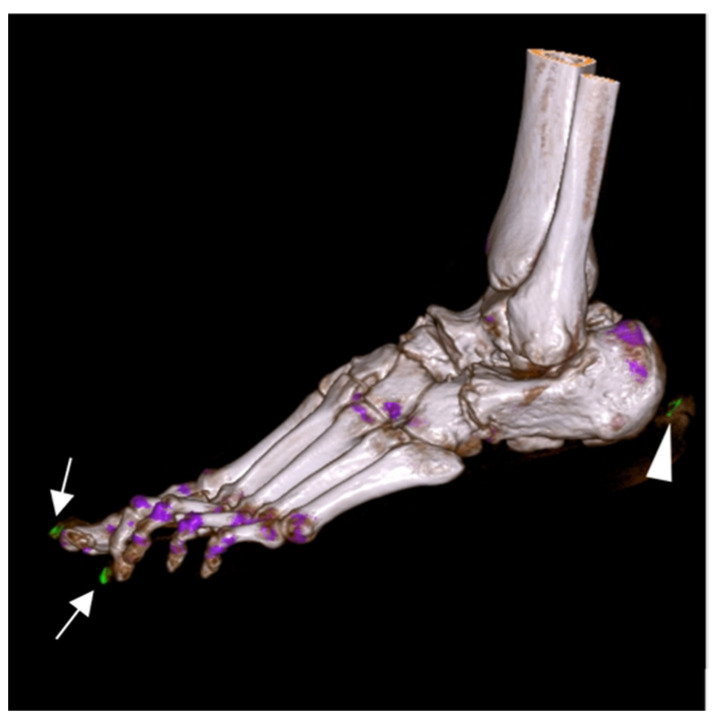
Dual-energy computed tomography of a foot with artifacts present in nailbeds (arrows) and skin (arrowhead).

**Figure 4 jcm-11-05249-f004:**
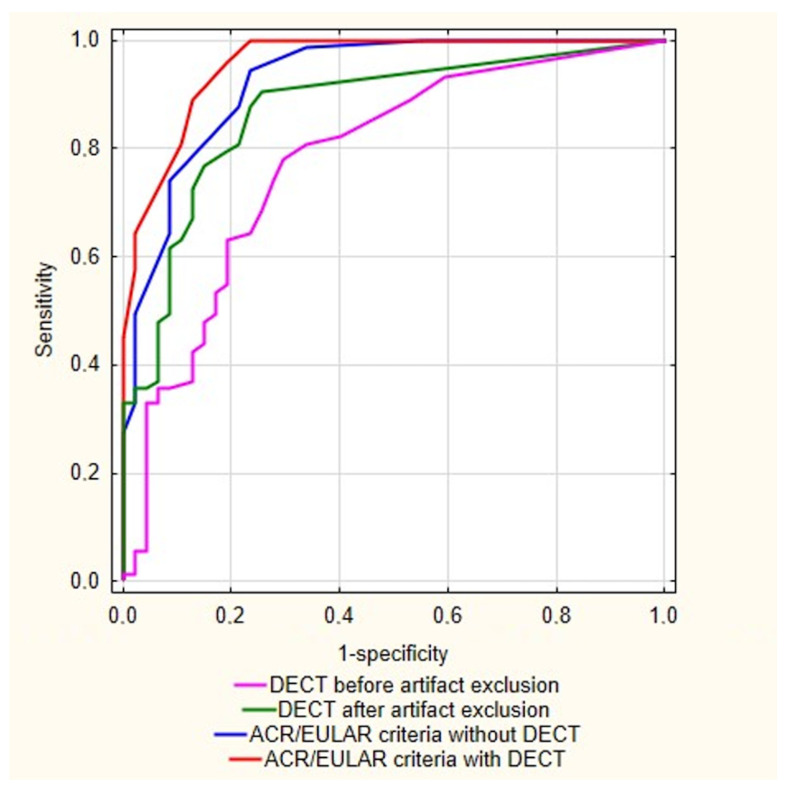
Receiver operating curves of the different gout diagnostic tools. DECT: dual-energy computed tomography, ACR: American College of Rheumatology, EULAR, European Alliance of Associations for Rheumatology.

**Table 1 jcm-11-05249-t001:** Anatomical areas examined by dual-energy computed tomography.

Anatomical Areas	Total	Positive for MSU Crystals
Hands	79	30 (38%)
Feet and ankles	141	95 (67%)
Knees	59	35 (59%)
Shoulders	6	5 (83%)
Elbows	33	15 (45%)
Total	318	180 (57%)

MSU: monosodium urate.

**Table 2 jcm-11-05249-t002:** Patients’ characteristics.

Patient Characteristics	Gout (*n* = 73)	Without Gout (*n* = 47)	Difference
Age (mean, ±SD)	55.4 (±12.1)	52.9 (±14.4)	ns
Sex—male (*n*, %)	61 (83.6%)	23 (48.9%)	*p* < 0.001
Obesity (*n*, %)	33(45.2%)	18(38.3%)	ns
Hypertension (*n*, %)	39 (53.4%)	26 (55.3%)	ns
Type 2 diabetes (*n*, %)	13 (17.8%)	7 (14.9%)	ns
Dyslipidemia (*n*, %)	40 (54.8%)	23 (48.9%)	ns
Kidney stones (*n*, %)	3 (4.1%)	3 (6.4%)	ns
Chronic kidney disease (*n*, %)	18 (24.7%)	8 (17%)	ns
Rheumatic conditions			
- Arthritis, rheumatoid, or unspecified (*n*, %)	7 (9.6%)	14 (29.8%)	*p* = 0.005
- Psoriasis or psoriatic arthritis (*n*, %)	6 (8.2%)	6 (12.8%)	ns
- Other spondyloarthritis (*n*, %)	10 (13.7%)	7 (14.9%)	ns
- Connective tissue disease (*n*, %)	4 (5.6%)	4 (8.5%)	ns
- Calcium pyrophosphate dihydrate deposition (*n*, %)	0	3 (6.4%)	-
- Infection arthritis (*n*, %)	1 (1.4%)	1 (2.1%)	ns
Hypothyroidism (*n*, %)	1 (1.4%)	4 (8.5%)	ns
Alcohol dependency (*n*, %)	1 (1.4%)	1 (2.1%)	ns
Myeloproliferative syndrome (*n*, %)	1 (1.4%)	0 (0%)	-

**Table 3 jcm-11-05249-t003:** Patients’ gout diagnostic features.

Gout Diagnostic Features	Gout (*n* = 73)	Without Gout (*n* = 47)	Difference
Uric acid in serum, mg/dL (mean, ±SD)	8.1 (±2.2)	6.1 (±2.3)	*p* < 0.001
Elevated uric acid in serum (*n*, %)	46 (63%)	16 (34%)	*p* = 0.002
Uric acid in 24 h urine collection, g/24 h (median, min, max)	0.46 (0.18, 1.2)	0.41 (0.29, 0.82)	ns
Excessive uric acid in urine (*n*, %)	4 (5.5%)	1 (2.1%)	ns
Tophus (*n*, %)	12 (16.4%)	0 (0%)	-
Features of gout in X-ray (*n*, %)	21 (28.8%)	3 (6.4%)	*p* = 0.003
Positive DECT result (*n*, %)	68 (93.2%)	28 (59.6%)	*p* < 0.001
- True deposits in DECT (*n*, %)	66 (90.4%)	12 (25.5%)	*p* < 0.001
- Artifacts in DECT (*n*, %)	48 (65.8%)	25 (53.2%)	ns
Diagnosis of gout according to ACR/EULAR criteria before DECT (*n*, %)	54 (74%)	4 (8.5%)	*p* < 0.001
Diagnosis of gout according to ACR/EULAR criteria after DECT (*n*, %)	73 (100%)	11 (23.4%)	*p* < 0.001
Uric-acid-lowering treatment (*n*, %)	41 (56.2%)	11 (23.4%)	*p* < 0.001

DECT: dual-energy computed tomography, ACR/EULAR: American College of Rheumatology/European Alliance of Associations for Rheumatology.

## Data Availability

The data underlying this article will be shared upon reasonable request to the corresponding author.

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
