# Peer review of "The Clinical Utility of Dual-Energy Computed Tomography in the Diagnosis of Gout—A Cross-Sectional Study"

_jcm, 2022, doi:10.3390/jcm11175249_

Round 1

Reviewer 1 Report

This is a very interesting study that evaluates the diagnostic performance of DECT alone and in combination with ACR/EULAR 2015 criteria in patients with suspected gout. The authors found better performance for the combination of ACR/EULAR criteria + DECT  (AUC = 0.95) compared to DECT alone (AUC = 0.92).

However, there are some major limitations due to its retrospective design and especially the absence of a "gold standard" with the detection of UMS crystals. The gold standard use by the authors is not clear. Furthermore, the relationship between MSU deposits volume and gout diagnosis need to be improved.

The diagnostic performance of DECT has been widely evaluated in the literature. This study shares similarities to the study of Gamala et al. who demonstrated that taking into account clinical + biological parameters (serum urate) was as effective as the combination of clinical + serum urate + DECT (AUC = 0.81 for both).

Regarding this study, we had several remarks:

Introduction

1)    Line 41-42: I would add that the search for MSU crystals can also be negative in case of extra-articular gout involvement.

2)    Line 58: DECT has been shown to be less effective in patients with a first ever gout attack and duration less than 24 months. I would change "in patients with a first ever gout attack" to "recent onset gout" and citation of Jia et al. (ref 17 in the paper)

Material and methods

3)    Line 70: Obesity is classically defined as a BMI > 30 kg/m2. Please correct

4)    An interesting data would be the time duration of the rheumatism. Is it possible to add this variable to the analysis? 

5)    Line 77: The authors mentioned the evidence of the double contour sign in the "data from imaging". However, this variable does not appear in the results. Either the proportion of patients with a DC sign should appear in table 2 or it should be stated that this variable was not evaluated.

6)    Line 80: please specify DECT settings (ratio, air and bone distance, HU min and max) and the voltage used (140/80 kV ?).

7)    Line 82: Was the DECT analysis performed blind to the clinic and biologics findings? if not, it is a major limitation.

8)    Line 86: "the final diagnosis of gout made by two rheumatologists": is this the gold standard used in this study? If so, please specify. What criteria did the authors use for the final diagnosis? Were they blinded to the results of the DECT?

Results

9)    Line 104: It would be clearer to say that 120 patients were enrolled in this study rather than 123 with 3 exclusions.

10) Figure 1: it should be mentioned that these are 3D reconstructions in the legend. Rather than projection, I will use the term "view".

11) Table 2:

a.     For patients with concomitant inflammatory rheumatism, was there evidence of gout (synovial fluid MSU crystals)?

b.     Please define "alcoholism".

c.     How do the authors explain that there are more artefacts in the "non-gout" group? 

d.     The authors find in ¼ of the non-gouty patients "true" MSU deposits. How can this presence of MSU deposits be explained?

e.     Please correct "according to EULAR criteria" to "according to ACR/EULAR criteria".

12) Line 133: did the optimal cut off point for MSU volume take into account DECT with artifact? If so, the authors need to recalculate the cut off value.

13) Does the cut off value of 0.05 allow differentiation between true and false MSU deposits?

14) Why do the authors mention costal cartilage when it is not one of the sites analyzed (Table 1)?

15) Figure 13: I think the sentence "the latter location could .... Deposits in subcutaneaous calcaneal bursa" is wrong. This is a classic "subcutaneous" artefact, probably related to hyperkeratosis at this level. I will remove this sentence.

16) Please explain the sentence "the best diagnostic value of DECT was obtained with any MSU volume". Indeed, sub-millimeter deposits are classically considered as artefacts.

Discussion

17) The authors can add the study of gamala et al. (DOI:10.1093/rheumatology/kez391) in the discussion.

18) Line 167: The relationship between deposit volume and gout diagnosis is not clear. Please clarify this relationship in the results (see comments 12-13)

19) Line 173 : some study have evaluated some post processing settings in order to reduce artifacts (park et al (10.2214/AJR.19.22222), strobl et al. (/10.2214/AJR.19.21404), dubief et al. (10.21037/qims-21-321). It could be discussed and add to references.

20) Line 178: I will be more moderate on this sentence. Because of the retrospective nature of this study, it seems difficult to determine whether a patient with ULT without MSU deposits on the DECT has never developed gout. It is possible that the treatment has dissolved the crystals which are no longer visible on the DECT.

21) Line 189: The authors report a significant proportion of false negative DECT results. However, in Table 2, a DECT positivity rate of 90% in the gout group is reported. Please clarify.

22) Line 193: I do not agree with this sentence. Deposits in CPPD are classically intra cartilaginous. Furthermore, the references used (2 and 18), recall this: “The "double contour "sign should be differentiated from hyperechoic foci within the substance of the cartilage, which is seen with calcium pyrophosphate dyhydrate deposition (CPPD)(ref 2).

The paper by Anjum et al (ref 18) does not support this claim (line 193) either.

Please correct

23) Line 196: This is a major limitation of this study. It is surprising that no synovial fluid results are available for this population.

Author Response

Thank you for the submitted reviews, which we read carefully and which we hope allowed us to improve our work. Please find below our responses to reviewers' comments.

Reviewer 2 Report

The current study (jcm-1872523) aims to assess the diagnostic value of DECT in gout diagnosis.

Overall comment: This study deals with an important subject (gout diagnosis using imaging studies). However, the current version of dataset, methods and their interpretation fell short of the expected scientific standard for diagnostic studies. The followings are the major concerns that must be addressed:

1. The reference method of gout diagnosis is not the gold standard. A diagnostic study should compare new diagnostic tool with the existing gold standard. In this case, a physician's clinical diagnosis of gout without MSU crystal identification was used as reference. This is not sufficient. This problem is even more relevant when we considered the high prevalence of rheumatic disease in the cohort (21 RA cases, 12 PsA cases, 17 SpA cases and 8 CPP arthritis cases). Many of these patients were 'clinically diagnosed with gout' but it is not possible to say for sure that they might just have flare of other rheumatic disease (e.g., PsA flare).

2. Inclusion criteria are not clear. What is 'people with clinical suspicion of gout'? Are they someone with active arthritis, history of joint pain, kidney stone or someone with undiagnosed subcutaneous nodules? Does having hyperuricemia alone count?

3. Methods are not clearly presented. Description of linear and logistic regression analysis is not clear - what are the dependent variables, what are the potential covariates, how were the covariate selected and which variables were adjusted for?

4. Some analysis does not match with the stated objectives. Logistic regression analysis of risk factor for gout is irrelevant to the primary question of the study (the diagnostic value of DECT). Analysis of the risk factors for gout is also not included in the introduction. 

5. The study design cannot support what the study claims to achieve. In the conclusion, the manuscript claims that DECT is helpful for gout diagnosis. This is not valid because the diagnostic values of DECT are compared against doctor's clinical judgement. The fact that DECT has sensitivity of 90% and specificity of 75% means that it is better to rely on doctor's diagnosis (i.e. the reference tool in this study). Furthermore, the study cannot claim that DECT improves the diagnostic value of the 2015 ACR/EULAR criteria because DECT is already part of that ACR/EULAR criteria.

Suggestions: I recommend revisiting the study protocol. The study may include only people hospitalized for 'arthritis' who underwent synovial exam for crystal and DECT. This setup would allow accurate assessment of DECT as a diagnostic tool for people admitted for arthritis, compared to the gold standard (crystal identification). I highly recommend adopting the STARD checklist for reporting diagnostic accuracy study (https://www.equator-network.org/reporting-guidelines/stard/).

Author Response

(The authors gave the same response as above.)

Round 2

Reviewer 1 Report

Thank you for this reviewed version. I have no further comments except :

- line 80 : "with" appears twice in the text.

- "risk factor for gout diagnosis" : i will remove "diagnosis" to the sentence.

Author Response

Thank you for submitted reviews. We found them very helpful and we are sure they allowed us to improve our work. Here are our responses.

- word "with" was deleted

- as requested, we removed the word "diagnosis" from the phrase "risk factors for gout diagnosis" 

Reviewer 2 Report

The authors have responded to all issues pointed out in the reviewers' reports. Though there were some unavoidable limitations to the dataset and design, the limitations are now clearly stated and discussed in the manuscript so readers could interpret the results with certain level of caution. Thank you for the opportunity the review the paper. I have no further comments.

Author Response

Thank you for your review and your comments which allowed us to improve our work.